# Pre-Clinical Investigation of Cardioprotective Beta-Blockers as a Therapeutic Strategy for Preeclampsia

**DOI:** 10.3390/jcm10153384

**Published:** 2021-07-30

**Authors:** Natalie K. Binder, Teresa M. MacDonald, Sally A. Beard, Natasha de Alwis, Stephen Tong, Tu’uhevaha J. Kaitu’u-Lino, Natalie J. Hannan

**Affiliations:** 1Translational Obstetrics Group, Department of Obstetrics and Gynaecology, University of Melbourne, Mercy Hospital for Women, Heidelberg 3084, Australia; nkbinder@unimelb.edu.au (N.K.B.); teresa.macdonald@mercy.com.au (T.M.M.); sally.beard@unimelb.edu.au (S.A.B.); maryd2@student.unimelb.edu.au (N.d.A.); stong@unimelb.edu.au (S.T.); t.klino@unimelb.edu.au (T.J.K.-L.); 2Therapeutics Discovery and Vascular Function Group, Department of Obstetrics and Gynaecology, University of Melbourne, Mercy Hospital for Women, Heidelberg 3084, Australia; 3Mercy Perinatal, Mercy Hospital for Women, Heidelberg 3084, Australia; 4Diagnostics Discovery and Reverse Translation, Department of Obstetrics and Gynaecology, University of Melbourne, Mercy Hospital for Women, Heidelberg 3084, Australia

**Keywords:** preeclampsia, beta-blocker, endothelial dysfunction, cardiovascular disease

## Abstract

Despite significant maternal and fetal morbidity, a treatment for preeclampsia currently remains an unmet need in clinical care. As too does the lifelong cardiovascular risks imparted on preeclampsia sufferers. Endothelial dysfunction and end-organ injury are synonymous with both preeclampsia and cardiovascular disease, including heart failure. We propose that beta-blockers, known to improve endothelial dysfunction in the treatment of cardiovascular disease, and specifically known to reduce mortality in the treatment of heart failure, may be beneficial in the treatment of preeclampsia. Here, we assessed whether the beta-blockers carvedilol, bisoprolol, and metoprolol could quench the release of anti-angiogenic factors, promote production of pro-angiogenic factors, reduce markers of inflammation, and reduce endothelial dysfunction using our in vitro pre-clinical preeclampsia models encompassing primary placental tissue and endothelial cells. Here, we show beta-blockers effected a modest reduction in secretion of anti-angiogenic soluble fms-like tyrosine kinase-1 and soluble endoglin and increased expression of pro-angiogenic placental growth factor, vascular endothelial growth factor and adrenomedullin in endothelial cells. Beta-blocker treatment mitigated inflammatory changes occurring after endothelial dysfunction and promoted cytoprotective antioxidant heme oxygenase-1. The positive effects of the beta-blockers were predominantly seen in endothelial cells, with a less consistent response seen in placental cells/tissue. In conclusion, beta-blockers show potential as a novel therapeutic approach in the treatment of preeclampsia and warrant further investigation.

## 1. Introduction

Preeclampsia is a serious pregnancy complication that affects 3–8% of all pregnancies. Because there are no effective medical therapies against the progression of preeclampsia aside from delivery, it remains a leading cause of maternal and perinatal deaths worldwide [1,2,3].

It is thought that preeclampsia develops after defective early trophoblast invasion and remodelling of the maternal spiral arterioles, causing significant oxidative stress [4,5]. Following this, excessive anti-angiogenic factors soluble fms-like tyrosine kinase-1 (sFlt-1) [6,7,8,9] and soluble endoglin (sENG) [10] are secreted into the maternal circulation [5,11,12,13]. These anti-angiogenic factors sequester circulating pro-angiogenic factors placental growth factor (PGF) and vascular endothelial growth factor (VEGF), reducing VEGF-mediated upregulation of endothelial nitric oxide [14]. Without an upregulation of endothelial nitric oxide, endothelin-1 (ET-1) production is likely uninhibited, and systemic inflammation and endothelial dysfunction ensues [15].

Endothelial dysfunction characterises preeclampsia and is a significant driver of the multi-organ injury clinically observed with the disease [16,17,18,19]. Circulating ET-1, a potent vasoconstrictor, is significantly increased in women with preeclampsia [20,21,22,23]. Increased ET-1 expression in the vasculature is also a marker of endothelial dysfunction. ET-1 receptor B (ETB) facilitates endothelium-mediated vasodilation by clearing ET-1 from circulation [24]. Vascular adhesion molecule (VCAM), another marker of endothelial dysfunction is also elevated with preeclampsia [25].

Considering the pathophysiology of preeclampsia, targeting the reduction of sFlt-1 and sENG production and release into the circulation has potential as a therapeutic strategy for the disease. A medical treatment that is safe in pregnancy, able to restore the angiogenic balance, improve endothelial function, and reduce inflammation, would likely prevent serious, long term damage to the maternal endothelium, and would represent a significant therapeutic advance. As such, we have developed a therapeutic testing pipeline to investigate whether existing drugs from other fields might be able to be repurposed to restore angiogenic balance and prevent endothelial dysfunction in preeclampsia [26,27,28,29,30,31,32].

Significantly, women who suffer preeclampsia are at increased risk of future cardiovascular disease, including a 4-fold increased risk of future heart failure [33]. Like preeclampsia, endothelial dysfunction and end-organ injury are also synonymous with heart failure and are associated with poorer prognosis [34,35,36,37,38]. The vasoactive peptide adrenomedullin (ADM) is a potent vasodilator with implications in both preeclampsia and cardiovascular disease [39,40]. In the management of heart failure, the beta-blockers carvedilol [41,42], bisoprolol [43], and metoprolol [44,45] have each been shown to reduce mortality to a similar extent [46]. As such, the use of any of them in treating patients with symptomatic heart failure constitutes standard therapy [47,48,49]. Beta-blockers while known to regulate blood pressure control, have also been shown to improve endothelial function when used in the treatment of cardiovascular disease [50]. While currently labetalol is the only beta-blocker used in the treatment of preeclampsia as an anti-hypertensive agent [51], in general many beta-blockers are considered safe in pregnancy based on their use in pregnant patients with cardiovascular disease [52]. The exception to this is atenolol, which is contraindicated given its association with small-for-gestational-age infants [52,53].

Given the relationship between preeclampsia, cardiovascular disease and heart failure at a pathophysiological level and in terms of subsequent lifetime risk, we hypothesised that the same beta-blockers able to modulate mortality risk in heart failure, might be of benefit in the treatment of preeclampsia. We therefore set out to evaluate the effects of carvedilol, bisoprolol, and metoprolol on the secretion of pro- and anti-angiogenic and inflammatory factors central to preeclampsia pathogenesis from placental and endothelial cells in vitro.

## 2. Materials and Methods

### 2.1. Tissue Collection

Ethical approval for this study was obtained from the Mercy Health Human Research Ethics Committee (R11/34). Women presenting to the Mercy Hospital for Women, Heidelberg, Australia, gave informed written consent for tissue collection. Placentas and umbilical cords were collected from normotensive term pregnancies (≥37 weeks’ gestation up to 41 weeks’ gestation) at elective caesarean section, where a fetus of normal customised birth weight centile was delivered. Samples were excluded where pregnancies were associated with gestational diabetes mellitus requiring insulin, preeclampsia or hypertension, congenital infection, chromosomal or congenital abnormalities, or evidence of chorioamnionitis (confirmed by placental histopathology) (Table 1). Samples were collected within 30 min of delivery and washed in cold phosphate-buffered saline (PBS).

### 2.2. Primary Human Umbilical Vein Endothelial Cell (HUVEC) Isolation

HUVECs were isolated from three or four individual umbilical cords per experiment, as previously described [54]. Briefly, the umbilical cord vein was cannulated and flushed with PBS to wash out blood cells. Next, 10 mL of collagenase (1 mg/mL, Worthington Biochemical Corporation, Lakewood, NJ, USA was infused into the cord and incubated at 37 °C for 10 min. The dissociated HUVECs were recovered by pelleting and resuspension, followed by culture in M199 media (Life Technologies, Carlsbad, CA, USA) containing 10% fetal calf serum (Thermo Fisher Scientific, Waltham, MA, USA), 1% antibiotic-antimycotic (Life Technologies), 1% endothelial cell growth factor (Sigma, St. Louis, Missouri, United States) and 1% heparin (Sigma). Cells were used between passage 2 to 4 and cultured at 37 °C in 20% O_2_ and 5% CO_2_.

### 2.3. Primary Human Cytotrophoblast Isolation

Human cytotrophoblasts were isolated from three or four individual placentas per experiment, as previously described [55]. Primary cytotrophoblasts were cultured in DMEM GlutaMAX (Life Technologies) containing 10% fetal calf serum and 1% antibiotic-antimycotic on fibronectin (10 mg/mL; BD Biosciences, Franklin Lakes, New Jersey, United States) coated wells. Cells were plated and allowed to attach over 12–18 h before washing with dPBS (Life Technologies) to remove cell debris. Cells were cultured under 8% O_2_, 5% CO_2_ at 37 °C.

### 2.4. Isolation and Culture of Placental Explants

Placental explants were isolated from three or four individual placentas per experiment. Small pieces of villous tissue were cut from the mid-portion of the placenta to avoid the maternal and fetal surfaces. These were thoroughly washed with PBS then dissected into small fragments of 1–2 mm size and three pieces put into each well of a 24 well plate. Explants were allowed to equilibrate at 37 °C for 12–18 h under 8% O_2_, 5% CO_2_ in DMEM GlutaMAX containing 10% fetal calf serum and 1% antibiotic-antimycotic.

### 2.5. Beta-Blockers In Vitro Experiments

Our in vitro models of preeclampsia recapitulate important characteristics of the disease pathogenesis, providing the opportunity to test therapeutic ability to target several key aspects of the disease, including excess secretion of anti-angiogenic factors (sFlt and sENG) and vascular endothelial dysfunction. These in vitro models of preeclampsia form the basis of our therapeutic testing pipeline [26,32,56,57].

Isolated primary HUVECs, cytotrophoblasts, and placental explants were treated with three different beta-blockers; carvedilol (1 uM and 10 uM, Sigma), bisoprolol (1 uM, 10 uM, 100 uM, and 1 mM, Sigma), and metoprolol tartrate (1 uM, 10 uM, and 100 uM, Sigma) in triplicate wells and cultured for 24 h. Media and cellular RNA were then collected and analysed by ELISA or quantitative RT-PCR (qPCR), respectively.

Endothelial dysfunction was induced in isolated primary HUVECs with TNFα (1 ng/mL, Life Technologies) for two hours prior to treatment with the beta-blockers (doses as above) and cultured for a further 24 h. Cellular RNA was collected and analysed by qPCR.

For HUVEC and cytotrophoblast experiments, a cell viability assay was run concurrently (CellTiter 96 AQueous Non-Radioactive Cell Proliferation Assay, Promega, Madison, WI, USA), according to manufacturer’s instructions.

### 2.6. ELISA

Concentrations of sFlt-1 and sENG were measured in HUVEC, cytotrophoblast and placental explant 24 h culture media using the DuoSet Human VEGF R1/Flt-1 and endoglin/CD105 ELISA kits, respectively (R&D Systems, Minneapolis, MN, USA). Optical density for ELISA was determined using a BioRad X-Mark microplate spectrophotometer (BioRad, Hercules, CA, USA), and protein concentrations calculated using BioRad Microplate manager 6 software.

### 2.7. Quantitative RT-PCR

Total RNA was extracted from isolated primary HUVECs, cytotrophoblasts and placental explants after drug treatment using the RNeasy mini kit (Qiagen, Valencia, CA, USA) and quantified using a NanoDrop ND 1000 spectrophotometer (NanoDrop Technologies Inc, Wilmington, DE, USA). RNA was converted to cDNA using the High-Capacity cDNA Reverse Transcription Kit (Applied Biosystems, Foster City, CA, USA) as per manufacturer guidelines. qPCR was performed using Taqman hydrolysis probes for *ENG* (Hs00923996_m1), *VCAM1* (Hs01003372_m1), *ET-1* (Hs00174961_m1), *NLRP3* (Hs00918082_m1), *IL1b* (Hs01555410_m1), *EDNRB* (Hs00240747_m1), *PTGS2* (Hs00153133_m1), *IL6* (Hs00985639_m1), *VEGF* (Hs00900055_m1), *PGF* (Hs00182176_m1), *HMOX1* (Hs01110250_m1) and *ADM* (Hs00181605_m1) on the CFX 384 (Biorad) using FAM-labelled Taqman universal PCR mastermix (Applied Biosystems) with the following run conditions: 50 °C for 2 min, 95 °C for 10 min, 95 °C for 15 s, 60 °C for 1 min (40 cycles). All data were normalized to a reference gene, *YWHAZ* (Hs01122454_m1), and the results graphed as fold change relative to control using the 2^−^^ΔΔCT^ method. The sFlt-1 splice variants *i13* and *e15a* were measured with Fast SYBR Green Master mix (Applied Biosystems) using primers specific for each variant as previously published [58], using *YHWAZ* as the reference gene with the following run conditions: 95 °C for 20 s, 95 °C for 1 s, 60 °C for 20 s (40 cycles). All samples were run in technical duplicate.

### 2.8. Statistical Analysis

All in vitro experiments were performed with technical triplicates and repeated a minimum of three times using tissue or cells isolated from different placentas.

Data was tested for normal distribution and statistically analysed as appropriate. When three or more groups were compared a 1-way ANOVA (for parametric data) or Kruskal–Wallis test (for non-parametric data) was used. Post hoc analysis was carried out using either the Tukey (parametric) or Dunn’s test (non-parametric). All data are expressed as mean ± SEM. *p* values < 0.05 were considered significant. Statistical analysis was performed using GraphPad Prism 8 software (GraphPad Software, La Jolla, CA, USA).

## 3. Results

### 3.1. Beta-Blocker Treatment Effects on the Secretion of Anti-Angiogenic Factors from Placental Cells/Tissue and Endothelial Cells

Treatment of primary HUVECs with the beta-blocker carvedilol significantly reduced the secretion of anti-angiogenic factor sFlt-1 (Figure 1A). The other two beta-blockers investigated, bisoprolol and metoprolol, did not affect sFlt-1 secretion from primary HUVECs. In primary human trophoblasts and placenta explants, all three beta-blockers had no effect on sFlt-1 secretion (Figure 1B,C, respectively).

Top dose metoprolol (1000 uM) resulted in a significant reduction in sENG secretion from primary HUVECs (Figure 1D), as did 100 uM bisoprolol in placental explants (Figure 1E).

### 3.2. Beta-Blocker Treatment Effects on the Expression of Anti-Angiogenic Factors in Placental Tissue and Endothelial Cells

In primary human placental explants, mRNA expression of the sFlt-1 isoforms, *sFlt-1-e15a* and *sFlt-1-i13* were not affected by treatment with any of the beta-blockers investigated (Figure 2A,B, respectively). Expression of *ENG* mRNA in primary HUVECs was significantly increased with top dose carvedilol (10 uM) and metoprolol (100 uM), but not bisoprolol at any dose (Figure 2C).

### 3.3. Beta-Blocker Treatment Effects on the Expression of Pro-Angiogenic Factors in Placental Tissue and Endothelial Cells

Expression of pro-angiogenic factors *VEGF*, *PGF*, and *ADM* were increased by different beta-blocker treatment of primary HUVECs. VEGF mRNA expression was upregulated by metoprolol (Figure 3A), *PGF* by carvedilol, bisoprolol and metoprolol (Figure 3C), and *ADM* by carvedilol and metoprolol (Figure 3E). Neither *VEGF* nor *PGF* mRNA expression was differentially regulated in primary human explants following beta-blocker treatment (Figure 3B,D, respectively).

### 3.4. Beta-Blocker Treatment Effects on the Expression of Inflammatory Mediators in Endothelial Cells

Expression of inflammatory mediator *IL-1b* in HUVECs was not affected by beta-blocker treatment (Figure 4A). Additionally, in primary cultured HUVECs, beta-blocker treatment with top doses of carvedilol, bisoprolol, and metoprolol significantly increased mRNA expression of *PTGS2* (Figure 4E).

When TNFα was added to HUVECs to induce endothelial dysfunction, *IL-1b* expression was significantly increased from control levels and decreased with doses of bisoprolol and metoprolol, but not carvedilol (Figure 4B). TNFα induced endothelial dysfunction also significantly increased mRNA levels of *IL-6* (Figure 4C), *NLRP3* (Figure 4D) and *PTGS2* (Figure 4F). Beta-blocker treatment with top dose carvedilol, bisoprolol, and metoprolol significantly decreased induced *IL-6* expression (Figure 4C). *NLRP3* mRNA expression was not affected by beta-blocker treatment (Figure 4D). *PTGS2* expression increased with top dose carvedilol, decreased with 100 uM bisoprolol, and was unaffected by metoprolol (Figure 4F).

### 3.5. Beta-Blocker Treatment Effects on Antioxidant HO-1

Expression of the antioxidant enzyme *HO-1* was significantly increased in HUVECs following top dose treatment with carvedilol, bisoprolol, and metoprolol (Figure 5).

### 3.6. Beta-Blocker Treatment Effects on Endothelial Dysfunction Markers

TNFα was used to induce endothelial dysfunction in primary cultured HUVECs. Compared to the no-TNFα control, TNFα significantly upregulated *VCAM* mRNA (Figure 6A). The top dose of each beta-blocker, carvedilol, bisoprolol, and metoprolol, significantly reduced *VCAM* mRNA expression (Figure 6A). In the same model of endothelial dysfunction, *ET-1* and its receptor, endothelin-1 receptor B (*ETB*), mRNA expression was not affected by beta-blocker treatment (Figure 6B,C, respectively).

## 4. Discussion

We report that beta-blockers that successfully reduce mortality in heart failure, also exert effects consistent with a reduction in endothelial dysfunction in models of preeclampsia. Through our suite of in vitro studies, carvedilol, bisoprolol, and metoprolol demonstrated a modest improvement in angiogenic balance, and significant improvement in various markers of vasoactivity, inflammation, and endothelial dysfunction. Treatments that address endothelial dysfunction in preeclampsia represent a significant knowledge gap in our pursuit of preeclampsia therapeutics. While beta-blockers have exhibited a positive effect on endothelial dysfunction in a cardiovascular disease setting, the actions of beta-blockers in gestational tissues presents novel findings.

Most therapeutics being investigated for the treatment of preeclampsia are concerned with lowering excessive placental secretion of anti-angiogenic factors sFlt-1 and sENG. Here, beta-blockers had no effect on placental (isolated trophoblasts and placental explant) secretion of sFlt-1, and only a modest reduction in sENG secretion from placental explants with bisoprolol. This was seen at both the protein and RNA level. There was also no change in the expression of pro-angiogenic *VEGF* or *PGF* in placental explants following beta-blocker treatment. In isolated primary HUVECs, beta-blockers did help restore the angiogenic balance somewhat. They decreased sFlt-1 (carvedilol) and sENG (metoprolol) secretion, and increased *VEGF* (metoprolol), *ADM* (carvedilol and metoprolol), and *PGF* (carvedilol, bisoprolol, and metoprolol), expression. This is significant given that PGF is almost ubiquitously low in cases of, and even preceding the development of, preeclampsia [59,60]. If angiogenic balance could be restored with such a therapy, this might offer an important step forward in the treatment of preeclampsia, especially as these drugs are also used for blood pressure control. Given they exert the additional benefit of reducing injury to the endothelium this could be an important consideration for post-partum therapies also.

Aside from restoring the angiogenic balance, decreasing inflammation is also desirable in a therapeutic to treat preeclampsia. In isolated primary HUVECs, the beta-blockers did not significantly reduce expression of the inflammatory mediator, interleukin (IL)-1b. However, importantly when disease was modelled whereby HUVEC were stimulated with TNFα to induce endothelial dysfunction (and a state of inflammation), beta-blockers were able to significantly reduce *IL-1b* (bisoprolol and metoprolol) and *IL-6* (carvedilol, bisoprolol, and metoprolol), but they had no effect on the critical regulator of the inflammasome, *NLRP3*. Interestingly, in isolated primary HUVECs all three beta-blockers, carvedilol, bisoprolol, and metoprolol increased expression of *PTGS2*, which encodes cyclooxygenase 2 (COX2), an enzyme largely recognised to be pro-inflammatory. However, there is significant evidence to show that COX2 is bimodal; with its first peak initially driving inflammation, and its second peak (almost 4-fold greater in magnitude) coinciding with a resolution in inflammation [61]. Compared to normotensive controls, COX2 is decreased in the placenta [62] and circulation [63] of women with preeclampsia. Further, therapeutic inhibition of COX2 is associated with adverse cardiac events [64] and delayed inflammation resolution [61], suggesting beta-blocker treatment, by increasing *PTGS2* expression may be both beneficial in the treatment of preeclampsia and also the long-term increased cardiovascular risks the disease imposes. However, this is less clear in the case of endothelial dysfunction, where following TNFα stimulation of HUVECs, carvedilol increased *PTGS2* expression while bisoprolol decreased it. Further investigation is warranted to elicit the mechanisms at play behind some of the different responses seen to the three different drugs throughout our experiments.

Looking further into the endothelial dysfunction model, all three beta-blockers significantly reduced expression of vascular cell adhesion molecule 1 (*VCAM*), a critical factor in the inflammatory process involved in the recruitment and migration of leukocytes [65], and marker of endothelial dysfunction. Unfortunately, beta-blockers did not reduce vasoconstrictor *ET-1* expression, nor upregulate its receptor, *ETB*, that acts as a vasodilator by removing ET-1 from circulation [24]. Given the potency of ET-1 as a vasoconstrictor, as well as being a biomarker of endothelial dysfunction and its intricate association with both preeclampsia and cardiovascular disease and heart failure, we expected that the cardio-protective beta-blockers may have reduced its expression. Importantly however, ADM, as well as having pro-angiogenic and anti-inflammatory effects, is also vasoactive, acting as a potent vasodilator and is significantly upregulated by both carvedilol and metoprolol. Interestingly, ADM is upregulated in instances of congestive heart failure and myocardial infarction, as a compensatory mechanism to protect the vasculature [66,67,68].

In addition, treatment with all three beta-blockers demonstrated an increase in the expression of the cytoprotective antioxidant enzyme heme-oxygenase-1 (*HO-1*). HO-1 is an important enzyme regulated by nuclear factor (erythroid-derived 2)-like 2 (NRF2), that as well as protecting cells from programmed cell death, also inhibits the pathogenesis of inflammatory disease [69]. Given the inflammatory nature of preeclampsia, upregulation of HO-1 is a valuable characteristic of any potential therapeutic.

Our findings suggest that there may be merit in evaluating the beta-blockers carvedilol, bisoprolol, and metoprolol further for their potential in treating or preventing preeclampsia; or in decreasing the lifelong cardiovascular risk incurred by women who suffer the disease. This study suggests that any therapeutic benefits that they display are more likely due to increased endothelial expression of the important pro-angiogenic factors *PGF*, *VEGF* and *ADM*, and through mitigation of inflammatory changes that occur subsequent to endothelial dysfunction, rather than through regulation of placental anti-angiogenic factor secretion, which was only modestly affected at best. Moreover, the effects of the beta-blockers are primarily seen in endothelial cells, with a less consistent response seen in placental tissue.

To our knowledge, this is the first study to investigate the effects of carvedilol, bisoprolol, and metoprolol on sFlt-1, sENG and PGF secretion and expression. Despite this novelty, our results in regard to the beta-blockers’ anti-inflammatory properties and reduction in endothelial dysfunction are supported by previous studies from the cardiovascular field in which the drugs have had comparisons made. While we saw many similarities in the responses to the beta-blocker treatments, some of the differences demonstrated may plausibly be due to the different types of beta-blockers that carvedilol, bisoprolol and metoprolol represent. Carvedilol, a third generation non-selective beta-blocker has been found to demonstrate improved endothelial function [70] through antioxidant effects [71,72]; but the molecular mechanisms still require clarification [71,73]. Unlike carvedilol, bisoprolol is a second generation beta-1 selective beta-blocker. Despite these differences, carvedilol has not been shown to have benefit over bisoprolol in lowering oxidant stress [38]. Like carvedilol, bisoprolol has been found to improve endothelial function when used in the context of cardiovascular diseases; these include hypertension and angina [74,75]. Metoprolol is similar to bisoprolol in that it is also a selective beta-1 receptor blocker. Metoprolol has also been shown to benefit endothelial function and to have antioxidant properties in cardiovascular disease [76,77]. Given that all three beta-blockers significantly increased *HO-1* expression in this model, we may infer that carvedilol, bisoprolol and metoprolol may all act to increase endogenous antioxidant defences and thus may be of benefit in the treatment of preeclampsia. Metoprolol is available as both an immediate release (metoprolol tartrate—used in this study) and sustained release (metoprolol succinate) formulation. While the active ingredient remains the same, further investigation of sustained release metoprolol succinate, which has been shown to reduce mortality in heart failure [44,45] would be of great interest.

A further consideration is that safety data in pregnancy would be required for all of these medications if they were to be used to treat preeclampsia. Currently all three medications are classified as TGA category C—much like labetalol, a beta-blocker frequently used to treat hypertension in pregnancy. Promisingly however, carvedilol and metoprolol are considered safe in pregnancy based on their use in pregnant women with other cardiovascular diseases including heart failure [52], and the only beta-blocker which is actively contraindicated in pregnancy is atenolol [52]; it has been associated with small-for-gestational-age infants [53].

In conclusion, we have presented evidence for beta-blockers as potential therapeutics for preeclampsia, with the idea that along with mitigating some of the hallmarks of endothelial dysfunction associated with preeclampsia, these benefits may extend to combating the increased lifelong cardiovascular risks to which these women are subjected. Further evaluation of beta-blockers using whole vessel myography as well as animal models of preeclampsia are required prior to clinical safety and efficacy studies.

## Figures and Tables

**Figure 1 jcm-10-03384-f001:**
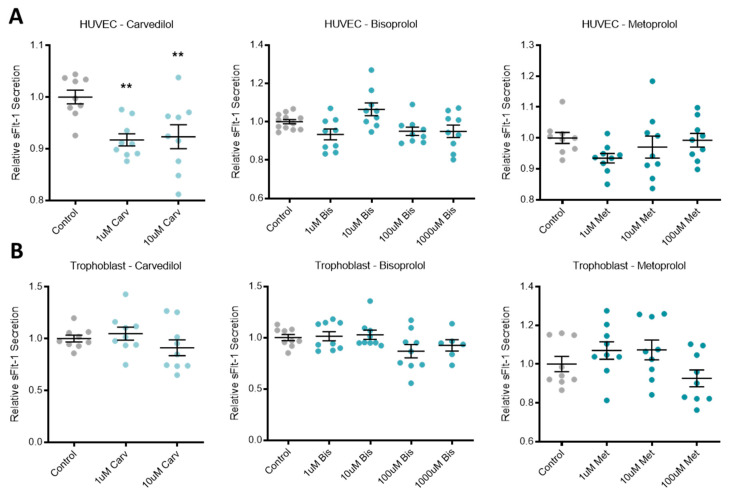
Beta-blockers had limited effect on secreted anti-angiogenic factors across different tissue types. Relative sFlt-1 secretion in HUVCEs (**A**) was significantly reduced with carvedilol (Carv) treatment, but not bisoprolol (Bis) treatment or metoprolol (Met) treatment. Relative sFlt-1 secretion was unchanged in trophoblasts (**B**) and explants (**C**) following beta-blocker treatment with Carv, Bis and Met. In HUVEC, relative sENG secretion (**D**) was decreased with treatment at the top dose of Met, but unchanged with Carv and Bis treatment. In explants, relative sENG secretion (**E**) was decreased with treatment at 100 uM Bis, but unchanged at the other doses or with Carv or Met. Data are mean ± SEM, expressed relative to control, *n* = 3–4, ** *p* < 0.01, **** *p* < 0.0001.

**Figure 2 jcm-10-03384-f002:**
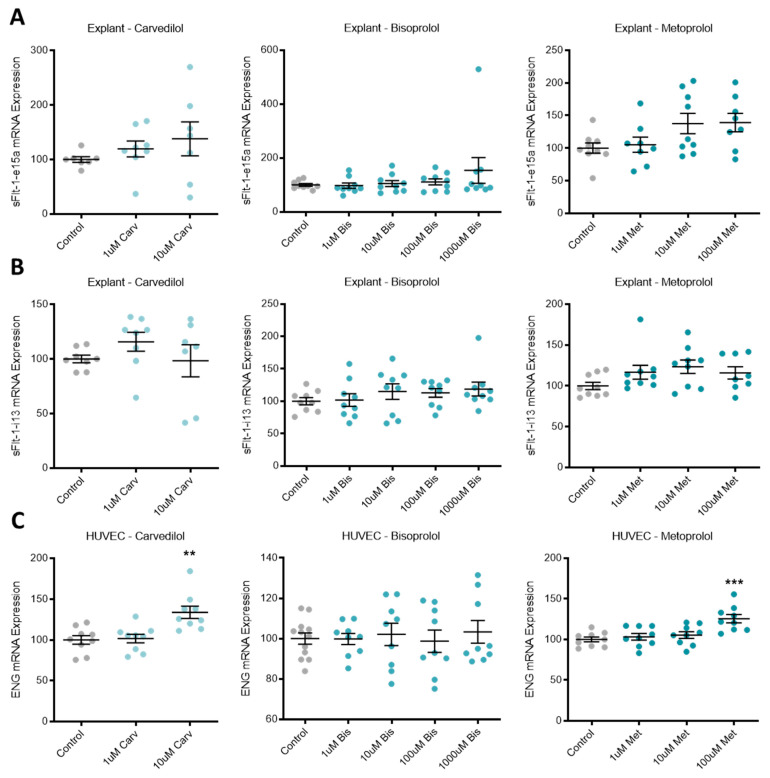
Endoglin mRNA is upregulated in HUVECs following beta-blocker treatment. Placental explant expression of sFlt-1 isoforms, sFlt-1-e15a (**A**) and sFlt-1-i13 (**B**) was not affected by treatment with beta-blockers carvedilol (Carv), bisoprolol (Bis), or metoprolol (Met). HUVEC expression of ENG mRNA (**C**) was significantly increased with top dose Carv and Met, but not Bis. Data are mean ± SEM, expressed as a percentage of control, *n* = 3–4, ** *p* < 0.01, *** *p* < 0.001.

**Figure 3 jcm-10-03384-f003:**
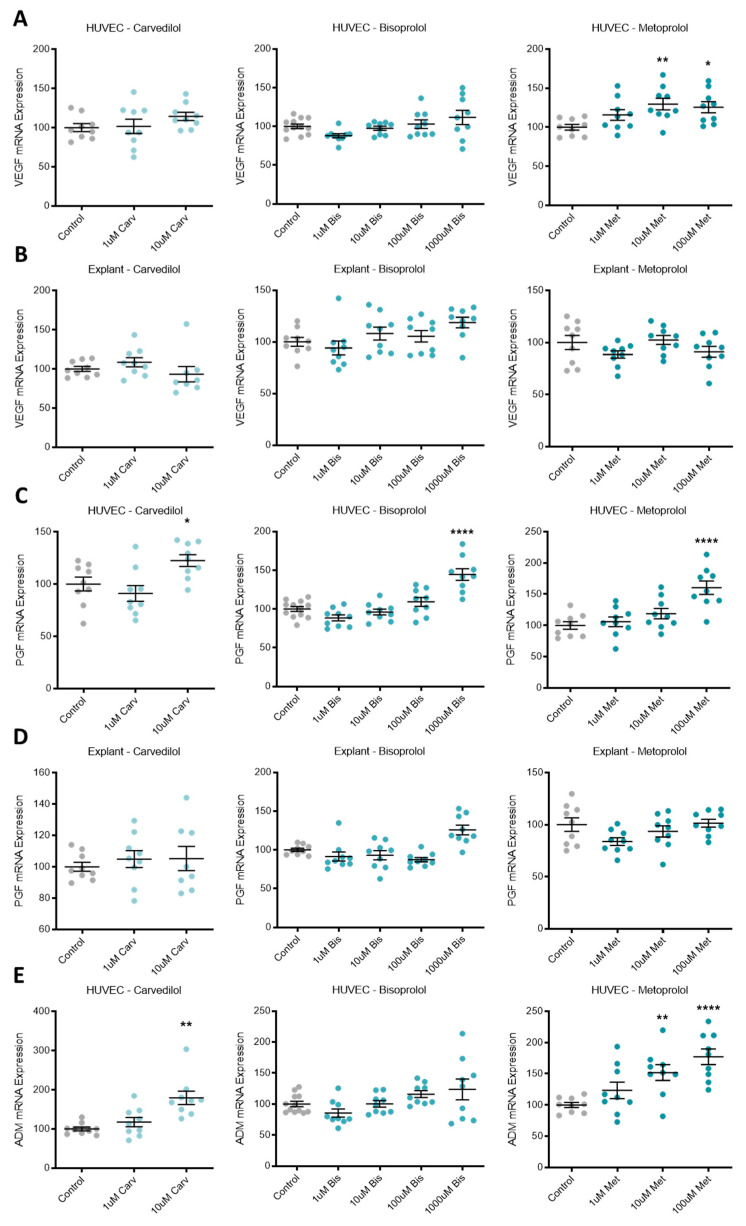
Pro-angiogenic factors upregulated following beta-blocker treatment of HUVECs, but not explants. VEGF mRNA expression in HUVECs (**A**) was significantly increased with metoprolol (Met) treatment, but not carvedilol (Carv) or bisoprolol (Bis). PGF mRNA expression in HUVECs (**C**) was significantly increased with the top dose of Carv, Bis, and Met. ADM mRNA expression in HUVECs (**E**) was significantly increased with top dose Carv treatment and Met treatment, but not Bis. VEGF and PGF mRNA expression were unchanged in placental explants following beta-blocker treatment (**B**,**D**, respectively). Data are mean ± SEM, expressed as a percentage of control, *n* = 3–4, * *p* < 0.05, ** *p* < 0.01, **** *p* < 0.0001.

**Figure 4 jcm-10-03384-f004:**
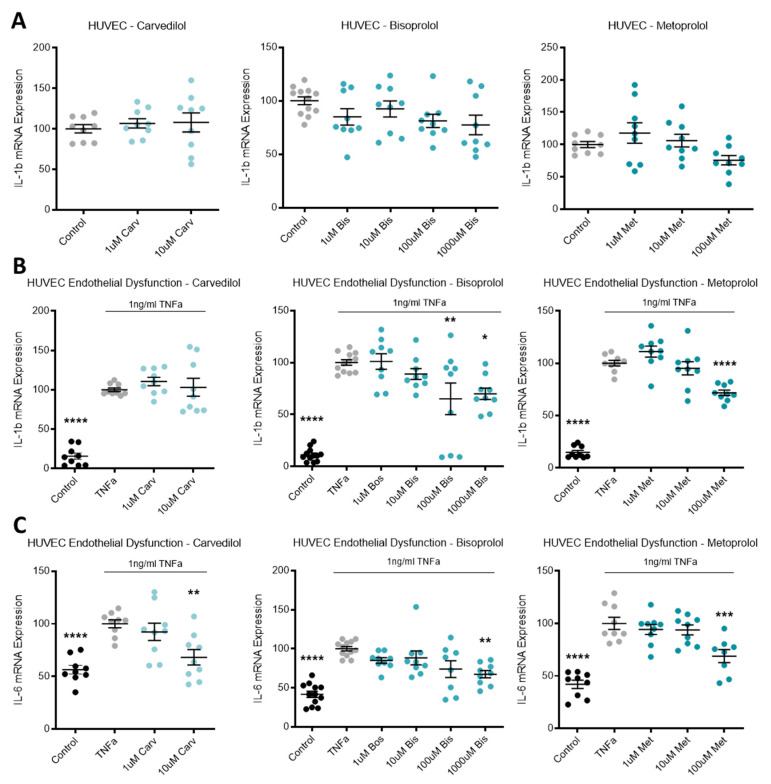
Inflammatory markers are altered following beta-blocker treatment. Expression of IL-1b mRNA in HUVECs (**A**) was not affected by treatment with beta-blockers carvedilol (Carv), bisoprolol (Bis), or metoprolol (Met). When TNFα is added to HUVECs to induce endothelial dysfunction, IL-1b expression is significantly increased from control levels (**B**) and decreases with doses of Bis and Met, but not Carv. TNFα induced endothelial dysfunction also significantly increases mRNA levels of IL-6 (**C**) and NLRP3 (**D**). Beta-blocker treatment with top dose Carv, Bis, and Met significantly decreases induced IL-6 expression (**C**). NLRP3 mRNA expression was not affected by beta-blockers treatment (**D**). Expression of PTGS2 mRNA in HUVECs (**E**) was significantly increased following top dose beta-blocker treatment with Carv, Bis, and Met. PTGS2 expression is significantly induced with TNFα (endothelial dysfunction) (**F**), which is further increased with top dose Carv, decreased with 100 uM Bis, and unaffected by Met. Data are mean ± SEM, expressed as a percentage of control (HUVEC) or TNFα control (endothelial dysfunction), *n* = 3–4, * *p* < 0.05, ** *p* < 0.01, *** *p* < 0.001, **** *p* < 0.0001.

**Figure 5 jcm-10-03384-f005:**
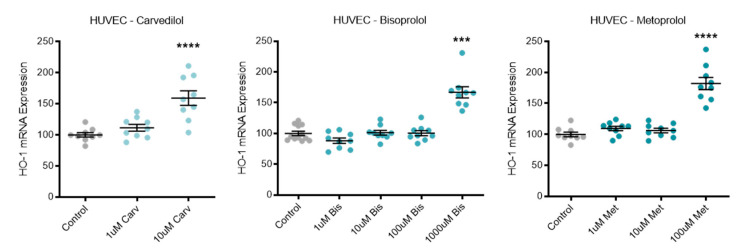
Antioxidant HO-1 is increased with beta-blocker treatment. Expression of HO-1 mRNA is significantly increased in HUVECs following top dose beta-blocker treatment with carvedilol (Carv), bisoprolol (Bis), and metoprolol (Met). Data are mean ± SEM, expressed as a percentage of control, *n* = 3–4, *** *p* < 0.001, **** *p* < 0.0001.

**Figure 6 jcm-10-03384-f006:**
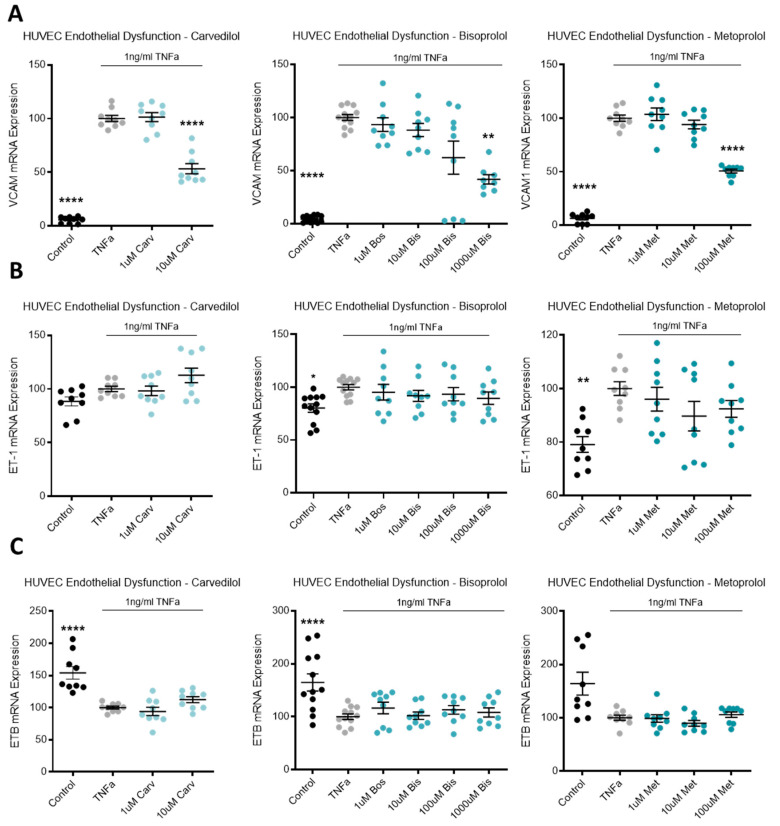
In a model of endothelial dysfunction, VCAM mRNA expression is decreased with beta-blocker treatment. In HUVECs treated with TNFα to induce endothelial dysfunction (**A**), VCAM mRNA is significantly reduced with top dose beta-blockers, carvedilol (Carv), bisoprolol (Bis), and metoprolol (Met). In the same model, ET-1 mRNA (**B**) and ETB mRNA (**C**) expression are not affected by Carv, Bis, or Met. Data are mean ± SEM, expressed as a percentage of TNFα control, *n* = 3–4, * *p* < 0.05, ** *p* < 0.01, **** *p* < 0.0001.

**Table 1 jcm-10-03384-t001:** Patient characteristics for term gestational tissue collection.

Characteristic	Number
Maternal Age, years (median (Q1, Q3))	33 (31, 38)
Fetal Sex (%)	
Male	8 (73)
Female	3 (27)
Maternal BMI (median (Q1, Q3))	22.7 (22.1, 25.4)
Smoker (%)	0
Birth Centile (%)	
<25th	0 (0)
26th–50th	2 (18)
51st–75th	6 (55)
76th–97th	3 (27)
>98th	0 (0)
Diabetes (%)	
None	10 (91)
GDM (diet)	1 (9)
Mode of delivery (%)	
Elective caesarean (not in labour)	11 (100)

## Data Availability

The data presented in this study are available on request from the corresponding author. The data are not publicly available due to patient privacy.

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
