# Peer review of "Pre-Clinical Investigation of Cardioprotective Beta-Blockers as a Therapeutic Strategy for Preeclampsia"

_jcm, 2021, doi:10.3390/jcm10153384_

Round 1
Reviewer 1 Report
I have only one question . Why the authors have not tested Labetalol? It is one of most often prescribed beta blocker in pregnancy hipertension and i feel missing this data
Author Response
Reviewer 1
I have only one question. Why the authors have not tested Labetalol? It is one of most often prescribed beta blocker in pregnancy hypertension and I feel missing this data.
RESPONSE: In this study, we specifically tested carvedilol, bisoprolol and metoprolol as these beta-blockers have mortality benefit when used in the treatment of heart failure. We took this approach because heart failure is a disease state that preeclampsia sufferers are at increased risk of developing, and which is also characterised by endothelial dysfunction. We did not test Labetalol, as it does not provide mortality benefit to heart failure patients, and unlike the other medications, would not represent a novel therapeutic for preeclampsia as it is already in use in clinical practice as an anti-hypertensive only.
In response to this comment, and to clarify we have made the following changes to the Abstract (page 1, lines 16-17 and 17-20):
“Endothelial dysfunction and end-organ injury are synonymous with both preeclampsia and cardiovascular disease”
Which now reads as:
“Endothelial dysfunction and end-organ injury are synonymous with both preeclampsia and cardiovascular disease, including heart failure”
and
“We propose that beta-blockers, known to improve endothelial dysfunction in the treatment of cardiovascular disease, may be beneficial in the treatment of preeclampsia”
Which now reads as:
“We propose that beta-blockers, known to improve endothelial dysfunction in the treatment of cardiovascular disease, and specifically known to reduce mortality in the treatment of heart failure, may be beneficial in the treatment of preeclampsia.”
We believe our hypothesis is clearly explained in the Introduction (page 2, lines 63-66, 68-73, and 78-81) when we state:
“Significantly, women who suffer preeclampsia are at increased risk of future cardiovascular disease, including a 4-fold increased risk of future heart failure [33]. Like preeclampsia, endothelial dysfunction and end-organ injury are also synonymous with heart failure and are associated with poorer prognosis [34-38].”
and
“In the management of heart failure, the beta-blockers carvedilol [41, 42], bisoprolol [43], and metoprolol XL [44, 45] have each been shown to reduce mortality to a similar extent [46]. As such, the use of any of them in treating patients with symptomatic heart failure constitutes standard therapy [47-49]. Beta-blockers while known to regulate blood pressure control, have also been shown to improve endothelial function when used in the treatment of cardiovascular disease [50].
and
Given the relationship between preeclampsia, cardiovascular disease and heart failure at a pathophysiological level and in terms of subsequent lifetime risk, we hypothesised that the same beta-blockers able to modulate mortality risk in heart failure, might be of benefit in the treatment of preeclampsia.”
In addition, our hypothesis and the reason for the choice of beta-blockers is again reemphasised in the first line of the Discussion (page 14, lines 280-282) which states:
“We report that beta-blockers that successfully reduce mortality in heart failure, also exert effects consistent with a reduction in endothelial dysfunction in models of preeclampsia.”
Reviewer 2 Report
The current study entitled: " Pre-clinical investigation of cardioprotective beta-blockers as a 2 therapeutic strategy for preeclampsia “has been revised.
Summary
In this paper authors hypothesized that the beta-blockers that are able to modulate mortality risk in heart failure, might be of benefit in the treatment of preeclampsia. They have created a pre-clinical in-vitro model in order to evaluate the effects of carvedilol, bisoprolol, and metoprolol on the secretion of pro- and anti-angiogenic and inflammatory factors central to preeclampsia pathogenesis from placental and endothelial cells that were collected from patients after delivery at term.
Results of the study suggests some potential therapeutic benefits for the treatment of beta-blockers that are more likely due to increased endothelial expression of the important pro-angiogenic factors PGF, VEGF and ADM, as well as through the mitigation of inflammatory changes, rather than through reduction of placental anti-angiogenic factor secretion.
Authors bring forward an important quarry regarding possible treatment option for preeclampsia which is a major cause of morbidity and mortality during and after pregnancy. I commend the authors for the in-vitro model that they have created, however, there seem to be some major concerns that were not addressed and make it almost impossible to conclude any pre-clinical as well as clinical conclusion for the readers. One such major issue is the fact that all three drugs that were used are classified as category C for pregnancy and breast-feeding (American an Australian) and so cannot be used to treat pregnant women.
Methodologically, the study is well designed, and results are clear.
Authors conclusions are clear, however, in my opinion results of the study mainly show the potential benefit of beta-blockers for the treatment of cardiovascular disease and not specifically preeclampsia and as such are not so innovative.
Conclusion:
Ultimately, I feel that the paper needs some major revisions and that the subject at matter would not be of interest to this journal.
Short summary of my comments:
Introduction:
- The introduction must include an explanation regarding the use of beta-blockers in pregnancy and their potential risks.
- Authors should elaborate as to the reason to choose the specific beta-blockers that were used.
Methods –
- Authors should elaborate regarding their in-vitro model and why it can be used for the evaluation of preeclampsia.
- Authors should provide information regarding the placental tissues that were used. Medical conditions such as diabetes mellitus might have an effect on the tissue cells that were obtained. More importantly, if these patients had any hypertensive disorder during pregnancy or chronic hypertension it might have major implications on results.
Results:
- A table with general characteristics of patients should be added.
- Section 3.1 - how does authors explain this lack of effect by two of the 3 drugs that were tested? is there a difference between carvedilol and the other two drugs that can explain these results? In addition, an explanation is needed for the different and opposit effect on sflt and sENG. This should be added to the discussion section.
- Section 3.2 - expression analysis does not support the results found for secretion, what is the explanation for that?
Discussion:
- First paragraph – “significant improvement in various markers of vasoactivity, inflammation, and endothelial dysfunction. This is perhaps not surprising, given beta-blockers’ normal function in the treatment of cardiovascular disease and blood pressure control”. Given this sentence, authors should better explain the innovation in their study
Author Response
Reviewer 2
In this paper authors hypothesized that the beta-blockers that are able to modulate mortality risk in heart failure, might be of benefit in the treatment of preeclampsia. They have created a pre-clinical in-vitro model in order to evaluate the effects of carvedilol, bisoprolol, and metoprolol on the secretion of pro- and anti-angiogenic and inflammatory factors central to preeclampsia pathogenesis from placental and endothelial cells that were collected from patients after delivery at term.
Results of the study suggests some potential therapeutic benefits for the treatment of beta-blockers that are more likely due to increased endothelial expression of the important pro-angiogenic factors PGF, VEGF and ADM, as well as through the mitigation of inflammatory changes, rather than through reduction of placental anti-angiogenic factor secretion.
RESPONSE: We thank the reviewer for this succinct and accurate summary of our study and its findings.
Authors bring forward an important quarry regarding possible treatment option for preeclampsia which is a major cause of morbidity and mortality during and after pregnancy. I commend the authors for the in-vitro model that they have created, however, there seem to be some major concerns that were not addressed and make it almost impossible to conclude any pre-clinical as well as clinical conclusion for the readers. One such major issue is the fact that all three drugs that were used are classified as category C for pregnancy and breast-feeding (American an Australian) and so cannot be used to treat pregnant women.
RESPONSE: We thank the reviewer for commending our model. With regard to the drugs being classified as Category C – yes this is the case, however the safety data available for some of the beta-blockers from women who have continued their use in pregnancy for cardiovascular disease, is very positive. We would point out that labetalol, a beta-blocker which is also currently used frequently as an anti-hypertensive in pregnancy, is also classified as Category C, and that recent expert review has classified both metoprolol and carvedilol as considered safe in pregnancy, just as labetalol is.
To address this issue in the manuscript we have added the following section to the Introduction (page 2, lines 73-77):
“While currently labetalol is the only beta-blocker used in the treatment of preeclampsia as a hypertensive agent [1], in general many beta-blockers are considered safe in pregnancy based on their use in pregnant patients with cardiovascular disease [2]. The exception to this Atenolol which is contraindicated given its association with small-for-gestational-age infants [2, 3].”
In addition, the following paragraph has also been added to the Discussion (page 16, lines 380-387):
“A further consideration is that safety data in pregnancy would be required for all of these medications if they were to be used to treat preeclampsia. Currently all three medications are classified as TGA category C – much like labetalol, a beta-blocker frequently used to treat hypertension in pregnancy. Promisingly however, carvedilol and metoprolol are considered safe in pregnancy based on their use in pregnant women with other cardiovascular diseases including heart failure [2], and the only beta-blocker which is actively contraindicated in pregnancy is atenolol [2]; it has been associated with small-for-gestational-age infants [3].”
Finally, we have added reference to clinical safety to the final line of the Discussion (page 16, lines 391-393):
“Further evaluation of beta-blockers using whole vessel myography as well as animal models of preeclampsia are required.”
now reads as:
“Further evaluation of beta-blockers using whole vessel myography as well as animal models of preeclampsia are required prior to clinical safety and efficacy studies.”
Methodologically, the study is well designed, and results are clear.
RESPONSE: We thank the reviewer for their comments.
Authors’ conclusions are clear, however, in my opinion results of the study mainly show the potential benefit of beta-blockers for the treatment of cardiovascular disease and not specifically preeclampsia and as such are not so innovative.
RESPONSE: Beta-blockers are undoubtedly beneficial in the treatment of cardiovascular disease. Importantly for this study, preeclampsia shares many common features with cardiovascular disease, namely endothelial dysfunction and end-organ injury, and women who have suffered from preeclampsia are at significantly increased lifelong risk of developing cardiovascular disease. The majority of therapeutic development in preeclampsia is focused on ablating placental secretion of anti-angiogenic factors sFlt and sENG. There is considerable need to address the cardiovascular aspects of preeclampsia too if we hope to both treat this complex disease, and mitigate the lifelong risk of cardiovascular disease in these women. This study specifically uses gestational tissues and in vitro models of preeclampsia to investigate beta-blockers in preeclampsia, which is a novel and innovative concept addressing an important knowledge gap.
To address this in the manuscript, the following change has been made to the Discussion (page 14, lines 284-288):
“This is perhaps not surprising, given beta-blockers’ normal function in the treatment of cardiovascular disease and blood pressure control”
now reads as:
“Treatments that address endothelial dysfunction in preeclampsia represent a significant knowledge gap in our pursuit of preeclampsia therapeutics. While beta-blockers have exhibited a positive effect on endothelial dysfunction in a cardiovascular disease setting, the actions of beta-blockers in gestational tissue presents novel findings.
Conclusion:
Ultimately, I feel that the paper needs some major revisions and that the subject at matter would not be of interest to this journal.
RESPONSE: Given the focus of the special issue is “Diagnostic or Therapeutic Strategies for Pregnancy Complications” I believe the work we have demonstrated here, using primary gestational human tissue to explore additional roles of these beta-blockers, and especially in models of endothelial dysfunction, that is central to the pathophysiology of preeclampsia is of interest. Thus we disagree that the subject matter would not be of interest to this journal.
Short summary of my comments
Introduction:
- The introduction must include an explanation regarding the use of beta-blockers in pregnancy and their potential risks.
RESPONSE: In response to this suggestion, the following statements have been added to the Introduction (page 2, lines 73-77):
“While currently labetalol is the only beta-blocker used in the treatment of preeclampsia as a hypertensive agent [1], in general many beta-blockers are considered safe in pregnancy based on their use in pregnant patients with cardiovascular disease [2]. The exception to this Atenolol which is contraindicated given its association with small-for-gestational-age infants [2, 3].”
- Authors should elaborate as to the reason to choose the specific beta-blockers that were used.
RESPONSE: In this study, we specifically tested carvedilol, bisoprolol and metoprolol as these beta-blockers have mortality benefit when used in the treatment of heart failure. We took this approach because heart failure is a disease state that preeclampsia sufferers are at increased risk of developing, and which is also characterised by endothelial dysfunction. We did not test Labetalol, as it does not provide mortality benefit to heart failure patients, and unlike the other medications, would not represent a novel therapeutic for preeclampsia as it is already in use in clinical practice as an anti-hypertensive only.
In response to this comment, and to clarify further we have made the following changes to the Abstract (page 1, lines 16-17 and 17-20):
“Endothelial dysfunction and end-organ injury are synonymous with both preeclampsia and cardiovascular disease”
now reads as:
“Endothelial dysfunction and end-organ injury are synonymous with both preeclampsia and cardiovascular disease, including heart failure”
and
“We propose that beta-blockers, known to improve endothelial dysfunction in the treatment of cardiovascular disease, may be beneficial in the treatment of preeclampsia”
now reads as:
“We propose that beta-blockers, known to improve endothelial dysfunction in the treatment of cardiovascular disease, and specifically known to reduce mortality in the treatment of heart failure, may be beneficial in the treatment of preeclampsia.”
We believe our hypothesis is clearly explained in the Introduction (page 2, lines 63-66, 68-73, and 78-81) when we state:
“Significantly, women who suffer preeclampsia are at increased risk of future cardiovascular disease, including a 4-fold increased risk of future heart failure [33]. Like preeclampsia, endothelial dysfunction and end-organ injury are also synonymous with heart failure and are associated with poorer prognosis [34-38].”
and
“In the management of heart failure, the beta-blockers carvedilol [41, 42], bisoprolol [43], and metoprolol XL [44, 45] have each been shown to reduce mortality to a similar extent [46]. As such, the use of any of them in treating patients with symptomatic heart failure constitutes standard therapy [47-49]. Beta-blockers while known to regulate blood pressure control, have also been shown to improve endothelial function when used in the treatment of cardiovascular disease [50].
and
Given the relationship between preeclampsia, cardiovascular disease and heart failure at a pathophysiological level and in terms of subsequent lifetime risk, we hypothesised that the same beta-blockers able to modulate mortality risk in heart failure, might be of benefit in the treatment of preeclampsia.”
In addition, our hypothesis and the reason for the choice of beta-blockers is again reemphasised in the first line of the Discussion (page 14, lines 280-282) which states:
“We report that beta-blockers that successfully reduce mortality in heart failure, also exert effects consistent with a reduction in endothelial dysfunction in models of preeclampsia.”
Methods:
- Authors should elaborate regarding their in-vitro model and why it can be used for the evaluation of preeclampsia.
RESPONSE: We are in a unique position to be able to collect precious primary human tissue samples from theatre (within two floors of our laboratory). This includes placenta and umbilical cord. From these primary tissues we can conduct both ex vivo placental tissue explant and isolation of cells for in vitro culture experiments.
Importantly our ex vivo and in vitro models simulate the key characteristics of preeclampsia including excess placental secretion of anti-angiogenic sFlt and sENG, and we model endothelial dysfunction, similarly to other fields but a key and important difference is we use endothelial cells obtained from pregnancy.
We have published many important findings using these models in over 20 publications in a number of high ranked Quartile 1 international journals see some listed below:
- Brownfoot, F., N. Binder, R. Hastie, A. Harper, S. Beard, L. Tuohey, E. Keenan, S. Tong and N. Hannan (2021). "Nicotinamide and its effects on endothelial dysfunction and secretion of antiangiogenic factors by primary human placental cells and tissues." Placenta 109: 28-31.
- Binder, N. K., F. C. Brownfoot, S. Beard, P. Cannon, T. V. Nguyen, S. Tong, T. J. Kaitu'u-Lino and N. J. Hannan (2020). "Esomeprazole and sulfasalazine in combination additively reduce sFlt-1 secretion and diminish endothelial dysfunction: potential for a combination treatment for preeclampsia." Pregnancy Hypertens 22: 86-92.
- Brownfoot, F. C., R. Hastie, N. J. Hannan, P. Cannon, T. V. Nguyen, L. Tuohey, C. Cluver, S. Tong and T. J. Kaitu'u-Lino (2020). "Combining metformin and sulfasalazine additively reduces the secretion of antiangiogenic factors from the placenta: Implications for the treatment of preeclampsia." Placenta 95: 78-83.
- de Alwis, N., S. Beard, Y. T. Mangwiro, N. K. Binder, T. J. Kaitu'u-Lino, F. C. Brownfoot, S. Tong and N. J. Hannan (2020). "Pravastatin as the statin of choice for reducing pre-eclampsia-associated endothelial dysfunction." Pregnancy Hypertens 20: 83-91.
- Brownfoot, F. C., N. J. Hannan, P. Cannon, V. Nguyen, R. Hastie, L. J. Parry, S. Senadheera, L. Tuohey, S. Tong and T. J. Kaitu'u-Lino (2019). "Sulfasalazine reduces placental secretion of antiangiogenic factors, up-regulates the secretion of placental growth factor and rescues endothelial dysfunction." EBioMedicine 41: 636-648.
- Brownfoot, F. C., S. Tong, N. J. Hannan, P. Cannon, V. Nguyen and T. J. Kaitu'u-Lino (2018). "Effect of sildenafil citrate on circulating levels of sFlt-1 in preeclampsia." Pregnancy Hypertens 13: 1-6.
- Kaitu'u-Lino, T. J., F. C. Brownfoot, S. Beard, P. Cannon, R. Hastie, T. V. Nguyen, N. K. Binder, S. Tong and N. J. Hannan (2018). "Combining metformin and esomeprazole is additive in reducing sFlt-1 secretion and decreasing endothelial dysfunction - implications for treating preeclampsia." PLoS One 13(2): e0188845.
- Hannan, N. J., N. K. Binder, S. Beard, T. V. Nguyen, T. J. Kaitu'u-Lino and S. Tong (2018). "Melatonin enhances antioxidant molecules in the placenta, reduces secretion of soluble fms-like tyrosine kinase 1 (sFLT) from primary trophoblast but does not rescue endothelial dysfunction: An evaluation of its potential to treat preeclampsia." PLoS One 13(4): e0187082.
- Brownfoot, F. C., R. Hastie, N. J. Hannan, P. Cannon, L. Tuohey, L. J. Parry, S. Senadheera, S. E. Illanes, T. J. Kaitu'u-Lino and S. Tong (2016). "Metformin as a prevention and treatment for preeclampsia: effects on soluble fms-like tyrosine kinase 1 and soluble endoglin secretion and endothelial dysfunction." Am J Obstet Gynecol 214(3): 356.e351-356.e315.
- Hannan, N. J., F. C. Brownfoot, P. Cannon, M. Deo, S. Beard, T. V. Nguyen, K. R. Palmer, S. Tong and T. J. Kaitu'u-Lino (2017). "Resveratrol inhibits release of soluble fms-like tyrosine kinase (sFlt-1) and soluble endoglin and improves vascular dysfunction - implications as a preeclampsia treatment." Sci Rep 7(1): 1819.
- Onda, K., S. Tong, S. Beard, N. Binder, M. Muto, S. N. Senadheera, L. Parry, M. Dilworth, L. Renshall, F. Brownfoot, R. Hastie, L. Tuohey, K. Palmer, T. Hirano, M. Ikawa, T. Kaitu'u-Lino and N. J. Hannan (2017). "Proton Pump Inhibitors Decrease Soluble fms-Like Tyrosine Kinase-1 and Soluble Endoglin Secretion, Decrease Hypertension, and Rescue Endothelial Dysfunction." Hypertension 69(3): 457-468.
- Hastie, R., S. Tong, N. J. Hannan, F. Brownfoot, P. Cannon and T. J. Kaitu'u-Lino (2017). "Epidermal Growth Factor Rescues Endothelial Dysfunction in Primary Human Tissues In Vitro." Reprod Sci 24(9): 1245-1252.
- Brownfoot, F. C., S. Tong, N. J. Hannan, R. Hastie, P. Cannon and T. J. Kaitu'u-Lino (2016). "Effects of simvastatin, rosuvastatin and pravastatin on soluble fms-like tyrosine kinase 1 (sFlt-1) and soluble endoglin (sENG) secretion from human umbilical vein endothelial cells, primary trophoblast cells and placenta." BMC Pregnancy Childbirth 16: 117.
- Ye, L., A. Gratton, N. J. Hannan, P. Cannon, M. Deo, K. R. Palmer, S. Tong, T. J. Kaitu'u-Lino and F. C. Brownfoot (2016). "Nuclear factor of activated T-cells (NFAT) regulates soluble fms-like tyrosine kinase-1 secretion (sFlt-1) from human placenta." Placenta 48: 110-118.
- Brownfoot, F. C., S. Tong, N. J. Hannan, N. K. Binder, S. P. Walker, P. Cannon, R. Hastie, K. Onda and T. J. Kaitu'u-Lino (2015). "Effects of Pravastatin on Human Placenta, Endothelium, and Women With Severe Preeclampsia." Hypertension 66(3): 687-697; discussion 445.
- Brownfoot, F. C., S. Tong, N. J. Hannan, R. Hastie, P. Cannon, L. Tuohey and T. J. Kaitu'u-Lino (2015). "YC-1 reduces placental sFlt-1 and soluble endoglin production and decreases endothelial dysfunction: A possible therapeutic for preeclampsia." Mol Cell Endocrinol 413: 202-208.
- Onda, K., S. Tong, A. Nakahara, M. Kondo, H. Monchusho, T. Hirano, T. Kaitu'u-Lino, S. Beard, N. Binder, L. Tuohey, F. Brownfoot and N. J. Hannan (2015). "Sofalcone upregulates the nuclear factor (erythroid-derived 2)-like 2/heme oxygenase-1 pathway, reduces soluble fms-like tyrosine kinase-1, and quenches endothelial dysfunction: potential therapeutic for preeclampsia." Hypertension 65(4): 855-862.
- Brownfoot, F. C., N. Hannan, K. Onda, S. Tong and T. Kaitu'u-Lino (2014). "Soluble endoglin production is upregulated by oxysterols but not quenched by pravastatin in primary placental and endothelial cells." Placenta 35(9): 724-731.
- Tong, S., T. J. Kaitu'u-Lino, K. Onda, S. Beard, R. Hastie, N. K. Binder, C. Cluver, L. Tuohey, C. Whitehead, F. Brownfoot, M. De Silva and N. J. Hannan (2015). "Heme Oxygenase-1 Is Not Decreased in Preeclamptic Placenta and Does Not Negatively Regulate Placental Soluble fms-Like Tyrosine Kinase-1 or Soluble Endoglin Secretion." Hypertension 66(5): 1073-1081.
Importantly these models and the above listed publications have been central to demonstrating the efficacy of therapeutics taken in to preeclampsia clinical trials (see below/several others under review currently).
- Cluver CA, Hannan NJ, van Papendorp E, Hiscock R, Beard S, Mol BW, Theron GB, Hall DR, Decloedt EH, Stander M, Adams KT, Rensburg M, Schubert P, Walker SP, Tong S. Esomeprazole to treat women with preterm preeclampsia: a randomized placebo controlled trial. Am J Obstet Gynecol. 2018 Oct;219(4):388.e1-388.e17.
- Cluver C, Walker SP, Mol BW, Hall D, Hiscock R, Brownfoot FC, Kaitu'u-Lino TJ, Tong S. A double blind, randomised, placebo-controlled trial to evaluate the efficacy of metformin to treat preterm pre-eclampsia (PI2 Trial): study protocol. BMJ Open . 2019 Apr 24;9(4):e025809. doi: 10.1136/bmjopen-2018-025809.
- Cluver CA, Walker SP, Mol BW, Theron GB, Hall DR, Hiscock R, Hannan N, Tong S. Double blind, randomised, placebo-controlled trial to evaluate the efficacy of esomeprazole to treat early onset pre-eclampsia (PIE Trial): a study protocol. BMJ Open. 2015 Oct 28;5(10):e008211. doi: 10.1136/bmjopen-2015-008211.
For further context the following section has been added to the Methods (page 3, lines 128-133) to add further context regarding the strength of our models of preeclampsia:
“In vitro and ex vivo models of preeclampsia recapitulate important characteristics of the disease pathogenesis and thus potential to test therapeutic ability to target several key aspects of disease: including excess secretion of anti-angiogenic factors (sFlt and sENG) and vascular endothelial dysfunction. Our in vitro/ex vivo models of preeclampsia form the basis of our therapeutic testing pipeline [4-7].”
- Authors should provide information regarding the placental tissues that were used. Medical conditions such as diabetes mellitus might have an effect on the tissue cells that were obtained. More importantly, if these patients had any hypertensive disorder during pregnancy or chronic hypertension it might have major implications on results.
RESPONSE: All gestational tissue samples used in this study (placentas and umbilical cords) were collected at term (37-41 weeks gestation) caesarean section, from women without hypertension and with an appropriately grown baby. The following change has been made to the Methods (page 2, lines 89-95) to clarify the careful selection of gestational tissue used:
“Placentas and umbilical cords were collected from normal term pregnancies (≥37 weeks’ gestation up to 41 weeks’ gestation) at elective caesarean section”
now reads as:
“Placentas and umbilical cords were collected from normotensive term pregnancies (≥37 weeks’ gestation up to 41 weeks’ gestation) at elective caesarean section, where a fetus of normal customized birth weight centile was delivered. Samples were excluded where pregnancies were associated with gestational diabetes mellitus requiring insulin, preeclampsia or hypertension, congenital infection, chromosomal or congenital abnormalities, or evidence of chorioamnionitis (confirmed by placental histopathology) (Table 1).”
Results:
- A table with general characteristics of patients should be added.
RESPONSE: As suggested, we have added a patient characteristics table to the Results sections (page 4, lines 184-185):
Table 1: patient characteristics for term gestational tissue collection
|
Characteristic |
Number |
|
Maternal Age, years (median (Q1, Q3)) |
33 (31, 38) |
|
Fetal Sex (%) |
|
|
· Male |
8 (73) |
|
· female |
3 (27) |
|
Maternal BMI (median (Q1, Q3)) |
22.7 (22.1, 25.4) |
|
Smoker (%) |
0 |
|
Birth Centile (%) |
|
|
· <25th |
0 (0) |
|
· 26th-50th |
2 (18) |
|
· 51st-75th |
6 (55) |
|
· 76th-97th |
3 (27) |
|
· >98th |
0 (0) |
|
Diabetes (%) |
|
|
· None |
10 (91) |
|
· GDM (diet) |
1 (9) |
|
Mode of delivery (%) |
|
|
· Elective caesarean (not in labour) |
11 (100) |
- Section 3.1 - how does authors explain this lack of effect by two of the 3 drugs that were tested? is there a difference between carvedilol and the other two drugs that can explain these results? In addition, an explanation is needed for the different and opposite effect on sflt and sENG. This should be added to the discussion section.
RESPONSE: It is possible that carvedilol may have different effects compared to bisoprolol and metoprolol because it is a non-selective beta blocker as opposed to bisoprolol and metoprolol’s beta-1 selectivity, however this is only speculative. Further studies would be required to confirm the mechanisms at play. Despite some of the differences between agents seen in our results, we are buoyed by the promising effects elicited from all three beta-blockers, including increased pro-angiogenic PGF, reduced inflammatory IL-6, reduced VCAM and increased antioxidant HO-1. Furthermore we have investigated other classes of drugs, and likewise various drug compounds within each class have varying potency and effects (see ref #’s 4, 11 and 13 from the above list of publications). We wish to note there were no opposite effects between the drugs.
In response to the reviewers comment, the following change has been made to the Discussion (page 15, lines 324-326 and page 16 lines 357-363):
“Further investigation is warranted.”
now reads as:
“Further investigation is warranted to elicit the mechanisms at play behind some of the different responses seen to the three different drugs throughout our experiments.”
and
“Despite this novelty, our results in regard to the beta-blockers’ anti-inflammatory properties and reduction in endothelial dysfunction are supported by previous studies from the cardiovascular field.”
now reads as:
“Despite this novelty, our results in regard to the beta-blockers’ anti-inflammatory properties and reduction in endothelial dysfunction are supported by previous studies from the cardiovascular field in which the drugs have had comparisons made. While we saw many similarities in the responses to beta-blocker treatments, some of the differences demonstrated may plausibly be due to the different types of beta-blockers that carvedilol, bisoprolol and metoprolol represent.”
- Section 3.2 - expression analysis does not support the results found for secretion, what is the explanation for that?
RESPONSE: Alterations in gene expression do not always reflect protein production, or protein secretion. There are many important aspects to consider whether there are transcriptional changes, translation to protein or whether there might be posttranslational modifications. Here we aimed to assess and report changes at both the transcript and protein secretory level. We were not surprised when the gene transcript changes were not reflected in protein secretion and vice versa.
Discussion:
- First paragraph – “significant improvement in various markers of vasoactivity, inflammation, and endothelial dysfunction. This is perhaps not surprising, given beta-blockers’ normal function in the treatment of cardiovascular disease and blood pressure control”. Given this sentence, authors should better explain the innovation in their study.
RESPONSE: While this is certainly understood by the cardiovascular field, the importance of this work is the use of primary human gestational tissue, to assess the effects of beta-blockers in a model of preeclampsia, using cells and tissues specific to pregnancy. This is the first study to assess these cardioprotective drugs in a model of preeclampsia. Offering knowledge gain for the field and an important first step forward to ensure beta-blockers are considered for their potential benefit in preeclampsia.
As requested we have added further context to the manuscript discussion, the following change has been made please see page 14, lines 293 - 297:
“This is perhaps not surprising, given beta-blockers’ normal function in the treatment of cardiovascular disease and blood pressure control”
now reads as:
“Treatments that address endothelial dysfunction in preeclampsia represent a significant knowledge gap in our pursuit of preeclampsia therapeutics. While beta-blockers have exhibited a positive effect on endothelial dysfunction in a cardiovascular disease setting, the actions of beta-blockers in gestational tissue presents novel findings.